# Determination of the Extraction, Physicochemical Characterization, and Digestibility of Sulfated Polysaccharides in Seaweed—*Porphyra haitanensis*

**DOI:** 10.3390/md18110539

**Published:** 2020-10-28

**Authors:** Mingshuang Dong, Yanhui Jiang, Chun Wang, Qian Yang, Xiaolu Jiang, Changliang Zhu

**Affiliations:** College of Food Science and Engineering, Ocean University of China, 5 Yushan Road, Qingdao 266003, China; MingsDong@163.com (M.D.); jiangyanhui@ouc.edu.cn (Y.J.); wangchun_95@163.com (C.W.); 21190711078@stu.ouc.edu.cn (Q.Y.); jiangxl@ouc.edu.cn (X.J.)

**Keywords:** *Porphyra haitanensis* polysaccharides, physicochemical characterization, vitro digestibility

## Abstract

The aim of the study was to extract *Porphyra haitanensis* polysaccharides (PHPs) using the water extraction and alcohol precipitation methods and explore their antioxidant activity and physicochemical properties. The single-factor and Box-Behnken response surface methodologies were used to optimize the extraction of polysaccharides from *Porphyra haitanensis*. Our results showed that the polysaccharide yield was as high as 20.48% with a raw material to water ratio of 0.04, and extraction time of 3 h at 80 °C. The extraction rate observed was similar to the actual extraction rate, thus proving the reliability of the optimization model. The extracted polysaccharides primarily consisted of galactose, glucose, and fucose in the molar ratio 76.2:2.1:1, respectively. The high performance gel permeation chromatography (HPGPC) results showed that the molecular weight of the PHPs obtained was 6.3 × 10^5^ Da, and the sulfate content was 2.7 mg/mL. Fourier infrared spectroscopy was used to analyze the functional groups and structures of the polysaccharides. The effect of concentration, temperature, and pH on the apparent viscosity of the PHPs solution were studied using rheology experiments, which revealed that PHPs were a “non-Newtonian fluid” with shear-thinning behavior. The viscosity of the PHPs gradually increased with increasing sugar concentration, and decreased with increasing temperature, acidity, and alkalinity. Detection of the antioxidant activity of OH*, DPPH*, and ABTS* revealed that the scavenging activity of ABTS* was higher than that of OH* and DPPH* in the concentration range of 1–5 mg/mL. In the experiments of simulating gastric juice and alpha amylase in vitro, it was found that PHPs can better resist digestion of alpha amylase, and have better resistance than fructooligosaccharide (FOS), so PHPs have potential prebiotic activity. These findings demonstrate the potential of PHPs for use in the food and cosmetic industries.

## 1. Introduction

*Porphyra,* a seaweed belonging to the Division Rhodophyta and Family Bangiaacea, is widely distributed in the intertidal waters between the cold and subtropical zones [1]. *Porphyra haitanensis* is one of the most economical seaweeds cultivated in China [2]; it is made up of proteins, sulfated polysaccharides, amino acids, and minerals. Sulfated polysaccharides, one of the primary active ingredients [3] in *Porphyra*, are made up of disaccharide units of alternating 3-linked β-d-galactosyl residues and 4-linked α-l-galactosyl residues, in addition to few 6-sulfate residues.

Recently, seaweed polysaccharides have gained importance owing to their antiviral, antioxidant, and antitumor activities [4,5]. *Porphyra haitanensis* polysaccharide (PHP) has attracted much attention owing to its composition, which endows it with antioxidant [6], antitumor [7], hypolipidemic [8], and antiviral [9] properties and other bioactivities. Zhang et al., [10] prepared a sulfated variant of the *Porphyra* polysaccharide and found that its antioxidant activity was significantly enhanced compared to that of the unmodified *Porphyra* polysaccharide. Additionally, Zhang [11] also proved that *Porphyra* polysaccharides act as scavengers of free radicals and antioxidants, thereby playing an important role in preventing oxidative damage in organisms. Moreover, sulfated polysaccharides of different molecular weights exhibit different antioxidant activities, the lower the molecular weight, the better the antioxidant activity [12]. Guiping Gong et al. [13] performed water extraction and alcohol precipitation (temperature: 80 °C, raw material to water ratio: 1:20 (*w/v*), and time: 1.5 h) and obtained a *Porphyra* yield of 3.8%, while Bilal Muhammad Khan [14] used a temperature of 90 °C, raw material to water ratio of 1:30 (*w/v*), and extraction time of 3 h, and obtained a yield of 3.3% *Porphyra*. Additionally, the viscosity of the polysaccharides obtained affect their rheological properties, polysaccharides can be used as excipients in new drug delivery systems [15] and as stabilizers for oil-in-water systems in emulsions [16], which is crucial for understanding the structure and potential functions of the polysaccharides. The relationship between intestinal microecological flora and human health has attracted more and more attention, which has led to an increase in the demand for prebiotics. The experiments have shown that undigested plant polysaccharides have excellent prebiotic activities, such as rapeseed polysaccharides [17], Chimonobambusa quadrangularis polysaccharides, and those in seaweeds like Himathalia elongata and Gigartina pistillata [18]. However, the rheological properties and in vitro simulated digestion studies are scarce for *Porphyra* polysaccharides.

In this study, water extraction and alcohol precipitation methods were used to optimize the extraction of polysaccharides from *Porphyra haitanensis*. Based on the response index of the PHPs, the extraction process was optimized using the response surface methodology. The scavenging ability of OH* free radicals, DPPH*, and ABTS* was measured and analyzed to characterize their antioxidant capacity. Additionally, the rheological properties and in vitro digestibility of the PHPs were also studied. The purpose of this study was to provide a theoretical basis for the development and utilization of *Porphyra haitanensis.*

## 2. Results and Discussion

### 2.1. Single-Factor Experiments for the Extraction of Porphyra Polysaccharides

#### 2.1.1. Effect of Extraction Temperature on the Yield of Polysaccharides

Extraction was performed at different temperatures (70, 80, 90, and 100 °C), keeping the extraction time and raw material to water ratio constant at 2 h and 1/25 g/mL, respectively. Generally, the solubility of polysaccharides increases with increasing temperature. Moreover, high temperature promotes the diffusion of polysaccharides from the cells [19]. As shown in the Figure 1a, at first the extraction rate increased and subsequently decreased with increasing temperature. However, no significant difference was observed between the extraction rates at 80 °C and 90 °C, since the polysaccharide structure is destroyed at very high temperatures, leading to their degradation. Therefore, 80 °C was chosen as the optimum temperature.

#### 2.1.2. Effect of Extraction Time on the Yield of Polysaccharides

Extraction was performed using different extraction times of 1, 2, 3, 4, and 5 h. The temperature was 80 °C, and the raw material to water ratio was 1/25 g/mL. As shown in the Figure 1b, the extraction rate first increased and subsequently decreased. The extraction rate at 3 h and 4 h was 21.087% and 21.400%, respectively. No significant difference was observed between the extraction rates at 3 and 4 h. Excess extraction time leads to polysaccharide degradation [20], and thus 3 h was chosen as the optimum time.

#### 2.1.3. Effect of the Raw Material to Water Ratio on the Yield of Polysaccharides

Different raw material to water ratios of 1/15, 1/20, 1/25, 1/30, and 1/35 g/mL were tested, keeping the extraction time and temperature constant at 2 h and 80 °C, respectively. As shown in Figure 1c, as the raw material to water ratio increased from 1/15 to 1/20 g/mL, the extraction rate decreased from 13.432% to 13.477%, respectively. However, when the raw material to water ratio was continuously increased to 1/25 and 1/35 g/mL, the yield decreased from 14.982% to 12.132%, respectively, since a high raw material to water ratio reduces the concentration and viscosity of the extraction solvent, thereby facilitating the dissolution of polysaccharide molecules in the water. Through this experiment, 1/25 g/mL was chosen as the optimum raw material to water ratio.

### 2.2. Optimization of Polysaccharide Extraction

The Box-Behnken design for response surface methodology was used to optimize the technological conditions in the water extraction process of PHPs. The primary aim of RSM was to efficiently identify the optimum values of independent variables to maximize responses [21]. Based on the aforementioned single-factor experiments, the raw material to water ratio (A), extraction time (B), and extraction temperature (C) were the independent variables, and the polysaccharide yield (Y) was the response value. A three-factor and three-level Box-Behnken design was applied to the surface optimization experiment. The central experiment point was repeated in three groups. The experimental data and processing are shown in Table 1, the experimental data were analyzed by performing multiple regression, and a second-order polynomial equation was obtained to describe the relationship between the variable and the response value:Y = 20.33−0.19A + 0.17B + 0.87C−0.24AB−0.61AC−0.10BC−1.25A^2^−2.08B^2^−1.58C^2^(1)(R^2^ = 0.9563)

Pareto analysis of variance (ANOVA) was also used to analyze the significance of the developed model equation. The obtained results showed that the developed model had acceptable F value, *p* value, correlation coefficient (R^2^), and adj-R^2^ values [22]. It can be seen from Table 2 that the selected model was highly significant (*p* < 0.0001). The *p* value of lack of fit was 0.2840 (*p* < 0.05), which is not significant; also, the difference between the model and the experimental value was small. The correlation coefficient was 0.9563 (*p* > 0.9), indicating that the model fits well. The value of adj-R^2^ was 0.8776, indicating that the model explains 87.76% of the change in the response value. In summary, the regression model had a good fit, showed small test errors, and could accurately analyze and predict the yield of the PHP extraction process.

It can also be seen from the F and *p* values in Table 2 that temperature has the greatest effect on the extraction yield of PHPs, followed by time and raw material to water ratio. The items BC, AC, AB, B, and A have no significant effect (*p* > 0.05). However, the effects of the other items were significant (*p* < 0.05). Among these, A^2^, B^2^, C^2^, and C are extremely significant (*p* < 0.01). Therefore, a simple linear relationship does not exist between the experimental factors and the response value.

### 2.3. Analysis of the Interaction Effects between Two Factors

The design principle and analysis of the response surface methodology was based on the Design Expert 8.0.6 software. The 3D stereogram of the response surface intuitively reflects the type of interaction between each factor, and the response value provides the basis for optimized production [23]. A steep slope on the 3D graph surface and densely integrated elliptical contour lines indicate that two factors interact greatly. However, a gentle slope and round contour lines indicate that the interaction between two factors is not significant.

According to Figure 2a–f, a change in the parameters results in different effects on the yield of PHP. As can be seen from Figure 2a,b, the influence of A (raw material to water ratio) and C (temperature) on Y (the yield) can be determined from the density of the contour lines. When A = 0.04, C is at the highest point, i.e., 80 °C. As shown in Figure 2c,d, the influence of A and B (extraction time) on Y can be determined from the elliptical contour. The effect was not significant when A = 0.04, and B reached the highest point, i.e., at 3 h. When the value of A exceeded 0.04 and that of B exceeded 3 h, the polysaccharide extraction rate decreased. From Figure 2e,f, it can be seen that B and C (temperature) versus Y has a curved 3D stereogram, and relatively sparse contour lines, indicating that its impact is not significant. The yield reaches the highest point when B = 3 h and C = 80 °C. By analysis of the regression equation, the best process for obtaining PHP (calculated based on the yield) must have a raw material to water ratio of 0.04, extraction time of 3 h, and extraction temperature of 80 °C. Under these conditions, the extraction rate of PHP was 20.48%.

The above finding was based on the results of the response surface analysis, and it was verified by performing three parallel experiments. The conditions used in the verification experiments were as follows: raw material to water ratio of 0.04, extraction time of 3 h, and extraction temperature of 80 °C. Under these conditions, the extraction yield obtained was 20.33% (±0.15), which is similar to the predicted value, thus proving the reliability of the experimental results. The extraction rate of PHPs using traditional water extraction and alcohol precipitation methods is less than 5% [24,25]. The extraction rate of polysaccharides obtained in this experiment was similar to that reported by Shu-YingXu et al., where the following ultrasonic microwave-assisted extraction conditions were used: extraction time of 30 min, extraction temperature of 80 °C, and liquid–solid ratio of 42 mL/g [26]. Therefore, our optimized methodology for the extraction of PHPs were very significant.

### 2.4. Physicochemical Properties of PHP

The physical and chemical characteristics of the PHPs extracted using the optimized experimental variables are presented in Table 3. The protein content was 0.056 g/L, purity was 85%, and the purity of analytical pure agar was 87.2%. The purity of the PHPs obtained conformed to the food-grade standards. Sulfate content was measured using the BaCl_2_ turbidimetric method. A single peak was displayed on the liquid phase spectrum, indicating that the polysaccharide was homogeneous. It had an average molecular weight of 6.3 × 10^5^ Da, which corresponds to the molecular weight range of food-grade polysaccharides, which range from 2.0 × 10^5^ Da to 8.0 × 10^5^ Da. The monosaccharide composition was determined using acid hydrolysis. LC analysis showed that most of the polysaccharides were primarily made up of galactose units, and a small amount of glucose and fucose. As seen in Figure 3a,b, the molar ratio of galactose, glucose, and fucose was 76.2:2.1:1. The differences observed in the chemical composition of the PHPs can be attributed to the discrepancy in the raw materials used.

### 2.5. FT-IR Spectra of PHPs

The FTIR spectrum of PHPs is shown in Figure 3c: 3441.64 cm^−1^ was the tensile vibration peak of O-H, the tensile vibration of the alkane C-H bond was observed at 2931.58 cm^−1^, the peak of polymer bound water was detected at 1647.30 cm^−1^, and the peak of tensile vibration at 1361 cm^−1^ was found to be sulfate, which corresponds to the physical and chemical index values detected above. There was an O-S-O tensile vibration peak at 1227.05 cm^−1^, the tensile vibration of the C-O-C bond was at 1155.51 cm^−1^, the peak at 1075.70 cm^−1^ was the position of the tensile vibration of the glycosidic bond, and the 772.55 cm^−1^ peak was the galactose C4-O-S that was formed by the tensile vibration, and 578.62 was the position of O=S=O (bending) [27]. The spectrum was consistent with the results obtained by Bilal Muhammad Khan et al. [14].

### 2.6. Rheological Analysis

#### 2.6.1. Effects of Concentration on the Apparent Viscosity of the PHPs Solution

The flow curve of the PHPs solution in Figure 4a shows that an increase in the shear rate results in a gradual decrease in viscosity, indicating that PHPs display a shear-thinning behavior. This reveals the pseudoplastic fluid properties of PHPs, such as shear thinning behavior, which generally occur in polymer solutions owing to their high molecular weight [28,29]. Additionally, the polysaccharide structure affects the intermolecular force, which in turn affects the apparent viscosity [30]. When the shear rate was higher than 30 s^−1^, at a certain value, the viscosity increased gradually with an increase in concentration. In addition, the effect of a 0.9% (*w/w*) concentrated sugar solution on viscosity was significantly higher than that of other groups, indicating the concentration dependence of the PHPs solution.

#### 2.6.2. Effects of pH on the Apparent Viscosity of the PHPs Solution

Figure 4b reveals that the PHPs showed shear-thinning behavior in all the pH values tested: pH = 3, 5, 7, 9, and 11. The viscosity of the polysaccharides decreased with increasing shear rate. The apparent viscosity was higher at pH 7 compared to that at other pH values. The apparent viscosity at different pH values showed the following trend: pH 7.0 > pH 9.0 > pH 5.0 > pH 3.0 > pH 11.0. Under both acidic and alkaline conditions, hydrogen bonds are broken, thereby leading to a decrease in the molecular weight of the polysaccharides, a conformational change in the molecular chain [31], and ultimately a decrease in the apparent viscosity.

#### 2.6.3. Effects of Temperature on the Apparent Viscosity of the PHPs Solution

The relationship between shear rate and apparent viscosity of the PHPs solution (at 0.5% (*w/w*), pH = 7, and temperature between 30 °C and 90 °C is shown in Figure 4c. As the shear rate increased, the apparent viscosity decreased. At a certain shear rate, the viscosity gradually decreased with increasing temperature. Previous studies have also reported similar findings [32]. Molecular movement is accelerated at high temperatures, leading to a weakening of the connections and interactions. Additionally, PHPs hydrolysis at high temperatures also reduces the viscosity. Owing to thermal movement, the polymer chains also untangle. The temperature characteristic plays an important role in food transportation and baking. This experiment aims to provide a certain basis for the application of PHPs. 

### 2.7. In Vitro Antioxidant Activity

The scavenging ability of hydroxyl free radicals is an indicator of their antioxidant activity. Hydroxyl radicals are generally produced in the metabolic process, and they promote oxidative damage. Antioxidants react with hydroxyl radicals and reduce them, thus inhibiting oxidation reactions [33]. DPPH* (diphenyl bitter hydrazine radical) is a stable free radical. Antioxidants donate electrons or hydrogen ions to DPPH* free radicals, thus inhibiting the propagation phase of lipid oxidation [34]. Interaction of an antioxidant with DPPH* changes its color from purple to yellow. The antioxidant ability of a substance is judged by this color change. ABTS*, on the other hand, is a blue water-soluble cationic radical. Antioxidants react with it and cause the solution to fade, and the antioxidant ability is judged based on this discoloration [35].

Dose-dependent clearance activity of polysaccharides was observed using DPPH*, OH*, and ABTS* clearance tests. Results showed that the dose dependence was more pronounced at higher concentrations. The clearance activity also depends on the electron transfer ability, molecular weight, water solubility, uronic acid content, glucoside bond, and the type of polysaccharide.

As shown in Figure 5a–c, the lowest DPPH*, OH*, and ABTS* clearance effects of 27.10%, 13.13%, and 37.99%, respectively, were observed at a concentration of 0.5 mg/mL, while the highest clearance activity, observed at a concentration of 0.9 mg/mL, was 47.58%, 37.32%, and 54.39%, respectively. The DPPH*, OH*, and ABTS* assays showed that the PHPs extracted using our method showed higher antioxidant activity compared to those extracted by Bilal Muhammad Khan et al. [14]. In the DPPH*, OH*, and ABTS* scavenging activity assays, Vc showed the highest scavenging activity (99.5%, 98.0%, and 99.0%, respectively). At a concentration of 0.7 and 0.9 mg/mL, the radical scavenging efficiency of ABTS* was greater compared to that of the OH* and DPPH* radicals. Collectively, PHPs scavenge the ABTS* free radicals more efficiently, followed by DPPH*, and OH* free radicals. The antioxidant activity of PHPs was similar to that of comfrey polysaccharides extracted by HongmeiShang et al. [36].

### 2.8. The Digestibility of PHPs by Artificial Human Gastric Juice

Figure 6a,b shows the resistance of PHPs to artificial gastric juice digestion. Taking fructooligosaccharide (FOS) as a reference prebiotic, with the increase of pH, the degree of hydrolysis gradually decreases, showing a higher resistance of PHPs than that of FOS. PHPs digested by artificial gastric juice showed higher acid resistance under the conditions of pH 1, 2, 3, 4, and 5 for 6 h, with values of the degree of hydrolysis of PHPs of 2.13%, 1.52%, 1.23%, 0.72%, and 0.46%, respectively, and values of the degree of hydrolysis of FOS of 2.67%, 2.00%, 1.35%, 1.09%, and 0.98%, respectively. The possible reason for this situation was that glycosidic bonds are more likely to break under acidic conditions; it also shows that PHPs can reach the gastrointestinal tract stably, and can be used as prebiotics for probiotics without being severely degraded. Under normal circumstances, the composition of carbohydrates affects their digestibility, and β bonds are more stable than α bonds [17].

### 2.9. Analysis of the Digestibility of PHPs by α-Amylase

In the range of pH 4–8, the digestion curve of PHPs resisting alpha amylase is shown in Figure 7a,b. When the pH was 7, there was a higher digestibility, and the order was: PH: 7 > 8 > 6 > 5 > 4. Moreover, the digestibility did not change after 4 h. When digested for 6 h under different pH values (according to the above order), the digestion order of PHPs was 5.60%, 4.19%, 1.90%, 1.09%, and 0.90%, and the digestibility of FOS was 5.45%, 5.05%, 4%, 2.25%, and 0.33%. Compared with FOS, PHPs showed better enzyme resistance and had higher stability to alpha amylase. The resistance to alpha amylase was the primary condition for screening PHPs as a prebiotic, so it can reach the part in the digestive system where the probiotics are used smoothly. These results are similar to those found for Mangifera pajang fibrous pulp and its polysaccharides by S.H. Al-Sheraji et al. [37].

## 3. Materials and Methods

### 3.1. Materials, Reagents, and Equipment

Haitanensis were cultured near the coast of Fujian province, China.

The following reagents were of analytical grade: 1,1-diphenyl-2-picry-hydrazyl (DPPH*), trichloroacetic acid (TCA), ascorbic acid, ABTS (2,2′-Azinobis-(3-ethylbenzthiazoline-6-sulphonate)), and standard sugars. The following equipment was used: a constant temperature bath oscillator (THZ-82A, Suzhou weir laboratory supplies co. LTD.), multi-purpose rotary evaporator (RE-52, shanghai, china), visible spectrophotometer (WFJ2000, Uniko instruments co. LTD.), rheometer (MCR101, Austria Anton pa co. LTD.), and FT-IR spectrum (4000–400 cm^−1^) of the ulvan was recorded using a Magna-IR560 spectrometer (Nicolet Instrument Corp., Madison, WI, USA) with a resolution of 4 cm^−1^. 

### 3.2. Extraction of PHPs

Distilled water was added to dried powder of *Porphyra haitanensis* to obtain a raw material to water ratio of 1:30 (*w/v*). Extraction was carried out at 80 °C for 2 h, followed by centrifugation at 8000 rpm for 10 min. The extract was concentrated by reduced-pressure distillation, and precipitation was performed by adding 3 times the volume of 95% ethanol and centrifuging at 4800 rpm for 10 min. Subsequently, freeze-drying was carried out to obtain PHPs. The polysaccharide yield was calculated using the following equation:(2)$$\;\mathrm{polysaccharide}\;\mathrm{yield}\;\left(\%\right)=\frac{\left(\mathrm{weight}\;\mathrm{of}\;\mathrm{polysaccharide}\;\mathrm{extract}\right)}{\left(\mathrm{weight}\;\mathrm{of}\;\mathrm{each}\;\mathrm{powder}\;\mathrm{sample}\right)}\times 100\%$$

### 3.3. Experimental Design of RSM

Based on the results of the single-factor experiment, the extraction time, temperature, and raw material to water ratio were chosen as the process parameters. Three factors and three levels were designed for the Box-Behnken response surface experiment. Three central points were selected, and a total of 15 different test combinations were used to test the factor levels. As shown in Table 4, the mathematical model is as follows [38]:(3)$$Y=\alpha_{0}+\sum_{i=1}^{3}\alpha_{i}\chi_{i}+\sum_{i=1}^{3}\partial_{\mathrm{ii}}\chi_{i}^{2}+\sum_{i<j}^{3}\partial_{ij}\chi_{i}\chi_{j}$$
where Y is the predicted response (extraction yield of polysaccharide); α_0_, α_i_, α_ii_, and α_ij_ are the regression coefficients for intercept, linear, quadratic, and interaction terms, respectively; and X_i_ and X_j_ are the independent variables.

### 3.4. Physicochemical Properties

#### 3.4.1. General Analytical Methods

The carbohydrate content and purity of the PHPs was determined using the phenol-sulfuric acid method [39], the sulfate content using the barium chloride-gelatin method [40], and the protein content using the Bradford’s procedure [41].

#### 3.4.2. PHPs Composition

Approximately 4 mg of the polysaccharide sample was hydrolyzed with 2 M trifluoroacetic acid at 105 °C. Trifluoroacetic acid was removed using methanol under reduced pressure (≥0.095 MPa), at 50 °C. The hydrolysate was then converted to an acetylated acetonitrile derivative according to the conversion procedure. Derivatization analysis was carried out using high-performance liquid chromatography (HPLC) (Agilent 1100 HPLC, ZORBAX Eclipse XDB-C18 separation column (4.6 × 250 mm, 5 µm, Agilent Technologies, Stockport, UK), which was used at an ambient temperature of 30 °C. PMP (1-phenyl-3-methyl-5-pyrazolone) derivatives were eluted with a mixture of 0.1 M phosphate buffer (pH 6.7) and acetonitrile at a ratio of 83:17 (*v/v*, %) and a flow rate of 1 mL/min, which was used to analyze the derivatives at 245 nm. Lactose (internal standard), rhamnose, arabinose, xylose, mannose, fucose, galactose, glucose, glucuronic acid, and galacturonic acid were also converted into acetylated aldehyde nitrile derivatives [42].

#### 3.4.3. Determination of Molecular Weight

The average molecular weight was measured using high performance gel permeation chromatography (HPGPC) [43]. A TSK-gel G4000SWXL column and a refractive index detector were used. The mobile phase was a 0.02 M phosphate buffer solution of pH 6.0. The injection volume was 20 μL, flow rate was 0.3 mL/min, and temperature (oven and detector) was 30 °C. Dextran was used to prepare standards of 1 kDa, 3.65 kDa, 5 kDa, 12 kDa, 21 kDa, 80 kDa, and 150 kDa for the preparation of a standard curve.

#### 3.4.4. Fourier Transform Infrared Spectroscopy (FT-IR)

The sample PHPs (1 mg) and KBr (100 mg) particles were mixed then ground and the tablet press was used to reduce the particle size to less than 5 mm (thin and transparent small discs) and then the samples were mounted on the Fourier transform infrared spectrometer (FT-IR) for infrared measurements of the spectrum in the range of 4000 to 400 cm^−1^ [44].

### 3.5. Rheological Properties

#### 3.5.1. Effect of Concentration on Apparent Viscosity

PHPs were dissolved in water and their concentrations were adjusted to 0.1%, 0.3%, 0.5%, 0.7%, and 0.9%. A MCR101 rheometer with a pp50 rotor was used. The temperature was set at 25 °C and the shear rate was 10–100 s^−1^.

#### 3.5.2. Effect of pH on Apparent Viscosity

The pH of the PHPs solutions (0.5% (*w/v*)) was adjusted using 0.03 M HCl and 0.03 M NaOH to obtain pH values of 3.0, 5.0, 7.0, 9.0, and 11.0. The rheometer parameters described in Section 3.5.1 were used for measurement.

#### 3.5.3. Effect of Temperature on Apparent Viscosity

Polysaccharide solutions with a concentration of 0.5% (*w/v*) were set at temperatures of 30 °C, 45 °C, 60 °C, 75 °C, and 90 °C. The samples were prepared as described in Section 3.5.1.

### 3.6. In Vitro Antioxidant Activity

#### 3.6.1. DPPH* Radical Scavenging Effect

Approximately 2 mL of DPPH* solution (2 × 10^−4^ M, configured with methanol) and 2 mL of polysaccharide solution or ascorbic acid (Vc) solution of different concentrations were added to a test tube. The reaction was shielded from light, and the tube was incubated at room temperature for 30 min. The absorbance of the samples was measured at a wavelength of 517 nm. The original solvent was used as a blank, and Vc was used as the positive control. The DPPH* radical scavenging effect was calculated using the following equation [45]:Scavenging effect (%) = [1 − (A_i_ − A_i0_)/A_0_] × 100%(4)
where A_i_ is 2 mL DPPH* + 2 mL sample solution; A_i0_ is 2 mL sample solution + 2 mL solvent; and A_0_ is 2 mL DPPH* + 2 mL solvent.

#### 3.6.2. Hydroxyl Free Radical (HO) Scavenging Effect

Around 1 mL of 8 mM H_2_O_2_ solution, 9 mM FeSO_4_ solution, and 1 mL of 9 mM salicylic acid-ethanol solution were added to each aliquot of 1 mL polysaccharide solution of different concentrations. Following incubation at 37 °C for 30 min, the absorbance was measured at a wavelength of 510 nm [13]. Vc was used as the positive control. The hydroxyl free radical scavenging effect was calculated using the following equation:Scavenging effect (%) = [A_0_ − (A_i_ − A_i0_)]/A_0_ × 100%(5)
where A_0_ is a blank control measured by using distilled water instead of a polysaccharide solution, A_i_ is the sample absorption value, and A_i0_ is the polysaccharide background absorption value.

#### 3.6.3. ABTS* Radical Scavenging Effect

The ABTS* working solution was obtained by mixing equal volumes of 7.4 mM ABTS* and 2.6 mM potassium persulfate solution and placing it in the dark for 12 to 16 h. The mixture was then diluted using phosphate buffer solution (pH 7.4) until the absorbance at 730 nm was 0.70 ± 0.02 [46].

Subsequently, 2.4 mL of ABTS* working solution and 0.6 mL of the sample solution were mixed evenly, and placed at room temperature for 6 min. The absorbance was measured at 730 nm, and Vc was used as the positive control. The ABTS radical scavenging effect was calculated using the following equation:Scavenging effect (%) = [1 − (A_i_ − A_i0_)/A_0_] × 100%(6)
where A_i_ is 2.4 mL ABTS* + 0.6 mL sample solution; A_i0_ is 2.4 mL sample solution + 0.6 mL solvent; and A_0_ is 2.4 mL ABTS* + 0.6 mL solvent.

### 3.7. Analysis of the Digestibility of PHPs by Artificial Human Gastric Juice

PHPs sample and positive controls by artificial human gastric juice were first identified by the following methods, described previously by al-Sheraji et al. (2012). Artificial human gastric juice was prepared by dissolving 0.2 g KCl, 8.0 g NaCl, 14.35 g NaH_2_PO_4_, 8.25 g Na_2_HPO_4_·2H_2_O, 0.18 g MgCl_2_·6H_2_O, and 0.1 g CaCl_2_·2H_2_O in 1000 mL deionized water and using 5 M HCl solution to adjust the pH to 1, 2, 3, 4, and 5. A mixture of 5.0 mL PHPs or FOS solution (10 mg/mL) and 5.0 mL artificial gastric juice (at each pH value) was prepared and incubated in a water bath (37 ± 1 °C) for 6 h. At 0, 0.5, 1, 2, 4, and 6 h of digestion, 2.0 mL of the reaction mixture was taken out and the reducing sugar and total sugar contents were estimated by the 3, 5-dinitrosalicylic acid (DNS) method [47] and phenol-sulfuric acid method [39]. The hydrolysis degree of the sample was calculated according to the following formula:(7)$$\;\mathrm{degree}\;\mathrm{of}\;\mathrm{hydrolysis}\;\left(\%\right)=\frac{\;\mathrm{reducing}\;\mathrm{sugar}\;\mathrm{released}\;}{\;\mathrm{total}\;\mathrm{sugar}\;-\;\mathrm{initial}\;\mathrm{reducing}\;\mathrm{sugar}\;}\times 100$$

The reducing sugar released was the difference between the reducing sugar after hydrolysis and the initial reducing sugar.

### 3.8. Analysis of the Digestibility of PHPs by α-Amylase

According to the authors of [48] and in line with other methods to determine the resistance of the PHPs sample to alpha amylase, the PHPs sample and FOS (positive control) were dissolved in sodium phosphate buffer solution (20 mM), received 10 mg/mL of the sample solution, 20 mg of alpha amylase, and 500 mL sodium phosphate buffer (20 mM) mixed in sodium chloride (6.7 mM). They were eventually pH adjusted to pH values of 4, 5, 6, 7, and 8 to give (2 unit/mL) alpha amylase solutions with different pH values. A mixture of PHPs solution (10 mg/mL, soluble in sodium phosphate buffer) and amylase solution (at each pH) with a volume ratio of 1:1 was prepared and the solution was incubated in a water bath (37 ± 1 °C) for 6 h. Samples (1 mL) were collected from the mixture for 0, 0.5, 1, 2, 4, and 6 h, and the hydrolysis degree of PHPs samples was determined by calculating the reducing sugar and total sugar content.

### 3.9. Statistical Analysis

All experiments were repeated thrice (*n* = 3) and the data were expressed as mean ± standard deviation. The Design-Expert software (version 8.0.6, Stat-Ease, Inc., Minneapolis, MN, USA) was used for the design as well as data analysis of RSM. The SPSS 18.0 software (SPSS Inc., Chicago, IL, USA) was used for the calculation of statistical significance, which was assessed using one way analysis of variance (ANOVA) followed by Duncan’s test. *p* < 0.05 was considered to be statistically significant.

## 4. Conclusions

Optimum conditions for the extraction of PHPs (in terms of yield) were obtained using the Box-Behnken response surface design: the raw material to water ratio was 0.04, time was 3 h, and temperature was 80 °C. Under these conditions, the extraction rate of PHPs was 20.33% (±0.15). Therefore, this extraction method has little effect on the structure and function of polysaccharides and high yield. The extracted PHPs primarily consisted of three types of monosaccharides, namely galactose, glucose, and fucose, in the molar ratio 76.2:2.1:1. The average molecular weight of the PHPs was 6.3 × 10^5^ Da. The molecular weight affects the antioxidant activity of the polysaccharides to a certain extent. Free radical scavenging analysis showed that the relatively strong antioxidant activity of PHPs in vitro functioned in a concentration-dependent manner. PHPs showed higher scavenging activity on ABTS* compared to that on OH* and DPPH*. Rheological experiments showed that PHPs were “non-Newtonian fluids” with a shear-thinning behavior. In the experiment of simulating gastric juice and alpha amylase in vitro, PHPs had better resistance than FOS, and it is predicted that polysaccharides have good prebiotic activity. Following purification, PHPs can be used as thickeners in the food processing industry. They also provide a certain basis for food transportation and baking. Owing to the good antioxidant activity of PHPs, they are potential agents for use in the pharmaceutical and cosmetic industries.

## Figures and Tables

**Figure 1 marinedrugs-18-00539-f001:**
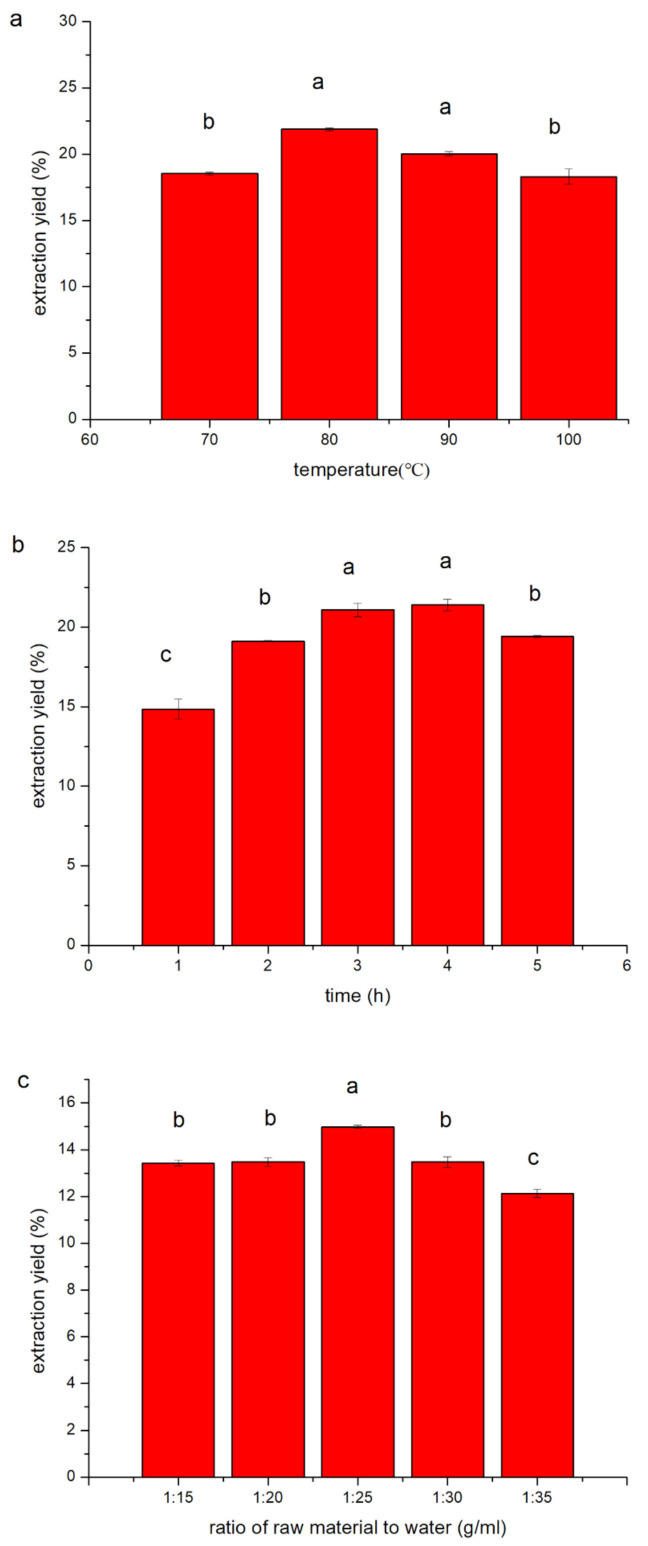
Effects of different factors on the yield of *Porphyra haitanensis* polysaccharides (PHPs). (**a**) Extraction temperature, (**b**) extraction time, (**c**) ratio of raw material to water. Data are means ± SD (*n* = 3), the error bars represent the standard deviation, values marked by the same letter are not significantly different (*p* < 0.05).

**Figure 2 marinedrugs-18-00539-f002:**
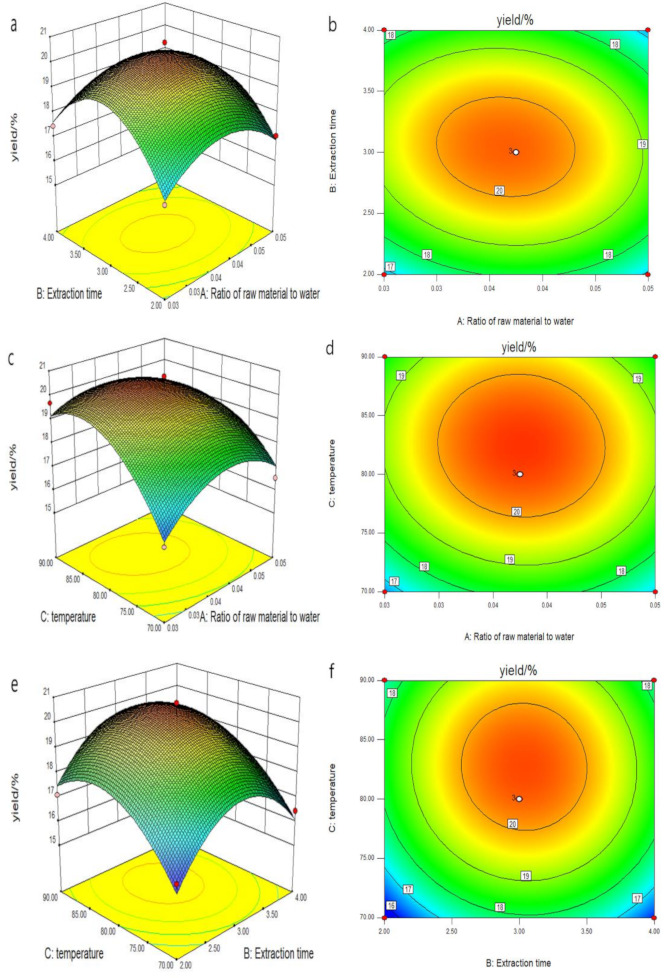
Figures of the variables’ mutual effects on the extraction yield. (**a****,b**) Response surface plots of A and B, (**c,d**) response surface plots of A and C, and (**e,f**) response surface plots of B and Cs. A, B, and C represent the ratio of raw material to water, extraction time, and temperature, respectively.

**Figure 3 marinedrugs-18-00539-f003:**
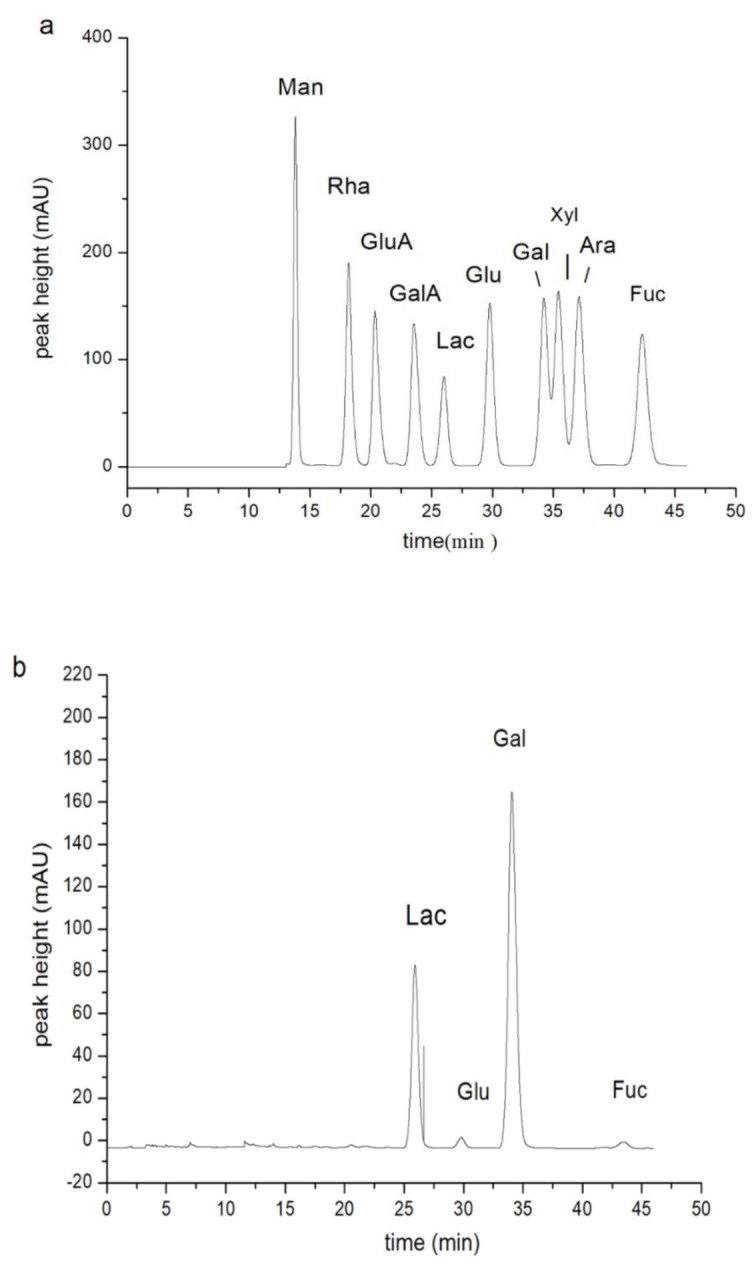
Monosaccharide components of a standard polysaaccharide (**a**), Monosaccharide components of a porphyra polysaccharide (**b**), FT-IR spectrum of PHPs (**c**).

**Figure 4 marinedrugs-18-00539-f004:**
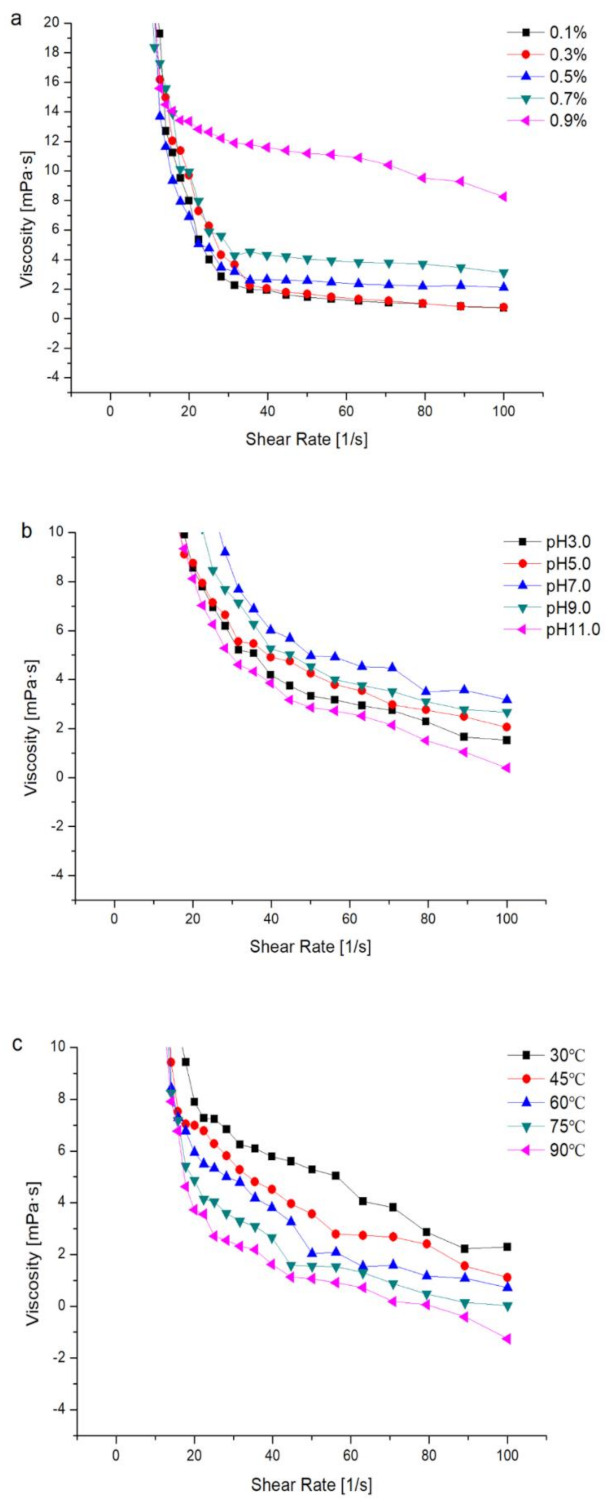
Apparent viscosity of different concentrations of porphyra polysaccharides with increasing shear rate (**a**). Apparent viscosity of porphyra polysaccharides (0.5%) with different pH values (**b**). Apparent viscosity of porphyra polysaccharides (0.5%) with different temperatures (**c**).

**Figure 5 marinedrugs-18-00539-f005:**
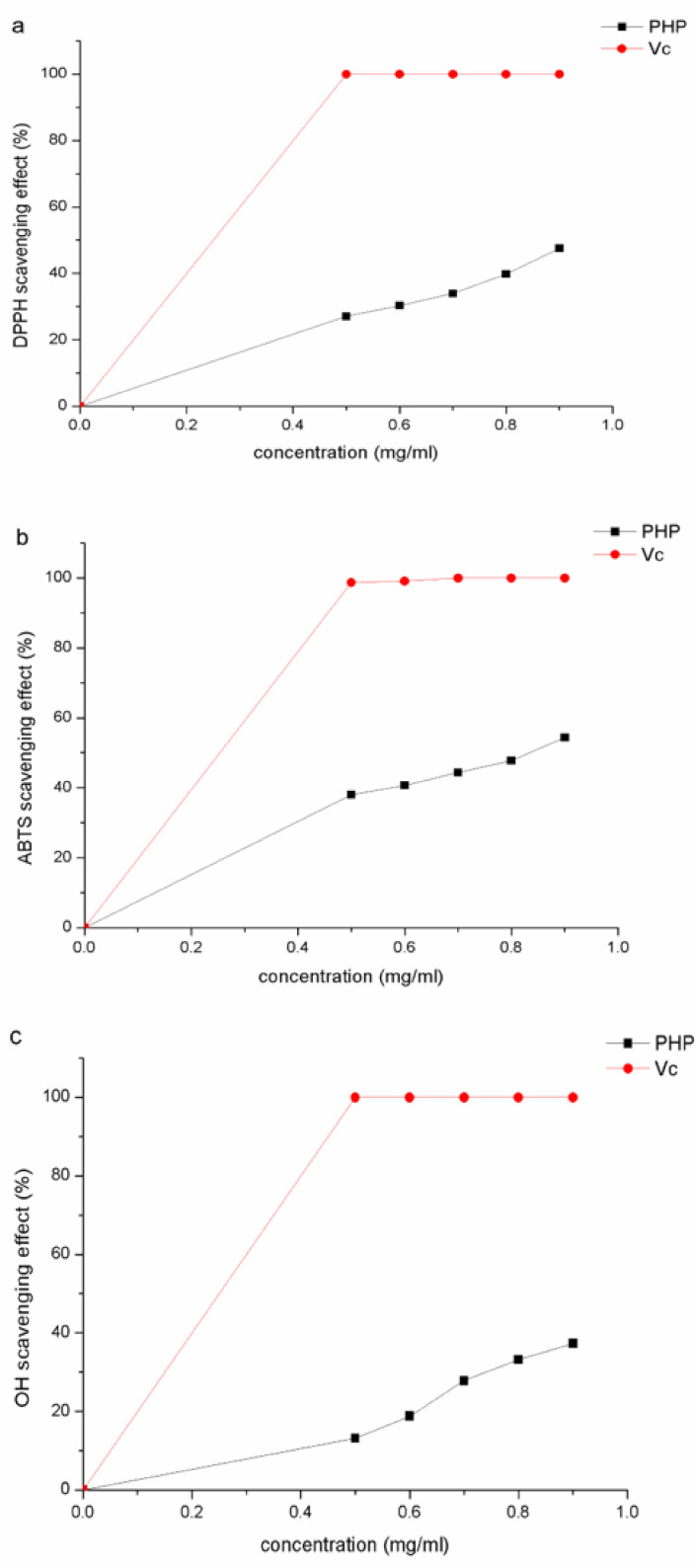
2,2-diphenyl-1-picrylhydrazyl (DPPH*) scavenging effect (**a**), 2,2′-azino-bis (ABTS*) scavenging effect (**b**), and hydroxyl radical (OH*) scavenging effect (**c**) of porphyra polysaccharides (mean ± SD, *n* = 3).

**Figure 6 marinedrugs-18-00539-f006:**
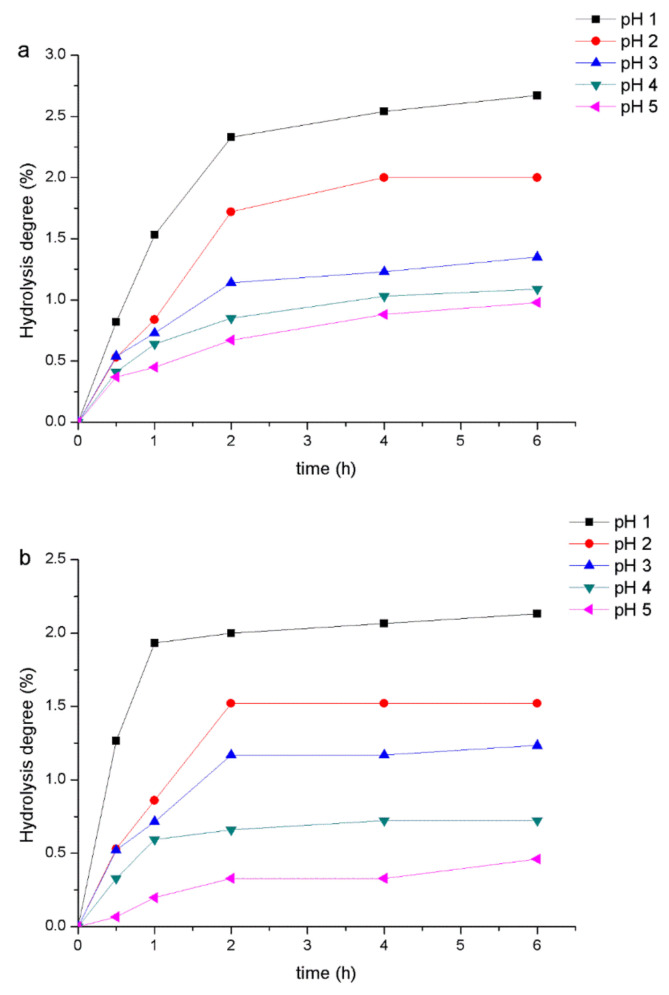
Effect of artificial human gastric juice on hydrolysis of fructooligosaccharide (FOS) (**a**) and PHPs (**b**).

**Figure 7 marinedrugs-18-00539-f007:**
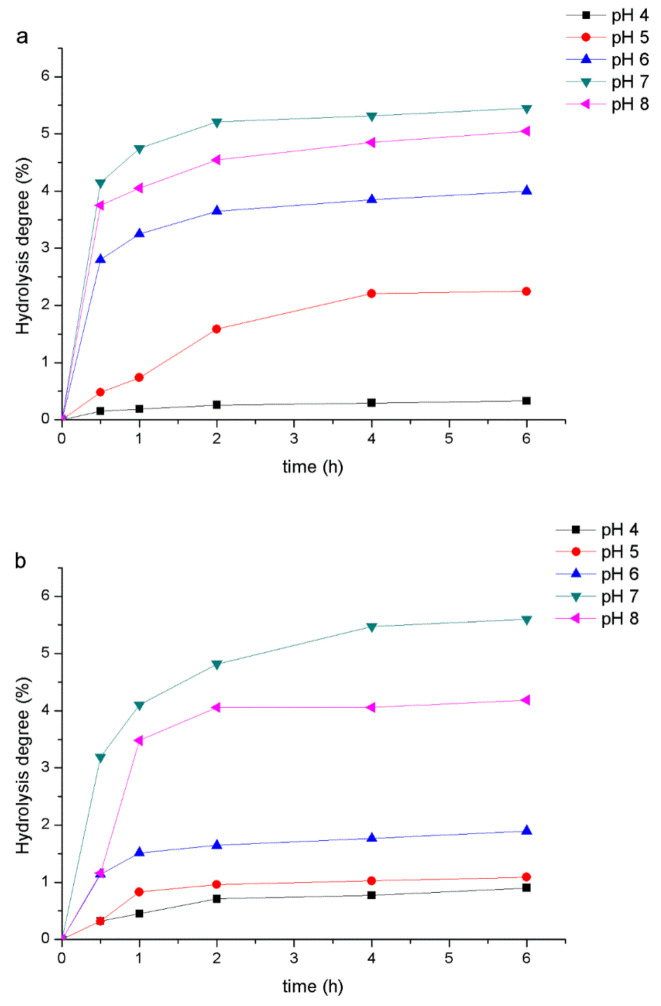
Effect of α-amylase on hydrolysis of FOS (**a**) and PHPs (**b**).

**Table 1 marinedrugs-18-00539-t001:** Box-Behnken test design and response values.

Number	A/(g/mL)	B/h	C/ °C	Y/%
1	0.05	3.00	90.00	17.79
2	0.03	4.00	80.00	17.45
3	0.04	4.00	90.00	17.24
4	0.04	3.00	80.00	20.05
5	0.05	2.00	80.00	17.05
6	0.03	3.00	70.00	16.01
7	0.04	2.00	70.00	15.9
8	0.04	2.00	90.00	17.1
9	0.04	3.00	80.00	20.79
10	0.04	3.00	80.00	20.15
11	0.03	3.00	90.00	19.69
12	0.05	4.00	80.00	16.91
13	0.05	3.00	70.00	16.53
14	0.04	4.00	70.00	16.45
15	0.03	2.00	80.00	16.61

**Table 2 marinedrugs-18-00539-t002:** Multivariate regression equation fitting for analysis of variance.

Source	Sum of Squares	df	Mean Square	*F* Value	*p* Value
model	35.32	9	3.92	12.15	0.0067
A	0.27	1	0.27	0.85	0.3995
B	0.24	1	0.24	0.75	0.4267
C	6.00	1	6.00	18.59	0.0076
AB	0.24	1	0.24	0.74	0.4280
AC	1.46	1	1.46	4.53	0.0865
BC	0.042	1	0.042	0.13	0.7331
A^2^	5.73	1	5.73	17.75	0.0084
B^2^	15.96	1	15.96	49.40	0.0009
C^2^	9.20	1	9.20	28.49	0.0031
Residual	1.62	5	0.32		
Lack of Fit	1.29	3	0.43	2.67	0.2840
Pure error	0.32	2	0.16		
Cor total	36.93	14			

Note: *p* < 0.05, significant difference; *p* < 0.01, the difference is very significant.

**Table 3 marinedrugs-18-00539-t003:** Characteristics of physicochemical properties from PHP.

Protein ^a^ Content/(g/L)	Polysaccharide Purity ^b^/%	Sulfate ^c^/(mg/mL)	Molecular Weight ^d^/Da	Monosaccharide Composition ^e^ (Molar Ratio)
				Glucose Galactose Fucose
0.056	85	2.7	6.3 × 10^5^	2.1 76.2 1

a: Evaluated by Coomassie Blue; b: Determination of sugar content by phenol-sulfuric acid method; c: Percentage of the weight of Polysaccharide (*v/v*); d: Evaluated by HPSEC as a component; e: The monosaccharide composition was detected by LC analysis (molar ratio).

**Table 4 marinedrugs-18-00539-t004:** Response surface experiment design factors.

Variable	Symbol		Level	
		−1	0	1
ratios of raw materials to water (g/mL)	X_1_	1:20	1:25	1:30
extraction temperature /°C	X_2_	70	80	90
extraction time/h	X_3_	2	3	4

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
