# Peer review of "Determination of the Extraction, Physicochemical Characterization, and Digestibility of Sulfated Polysaccharides in Seaweed—Porphyra haitanensis"

_marinedrugs, 2020, doi:10.3390/md18110539_

Round 1

Reviewer 1 Report

Determination of the extraction, physicochemical characterization, and digestibility of polysaccharides from Porphyra haitanensis.

This research article is about optimization of the extraction properties of the Porphyra haitanensis polysaccharides (PHP) using the water and alcohol precipitation methods. It was done single-factor experiment, the influence of extraction temperature, time, and the ratio of raw material to water ratio on the yield of polysaccharides were examined during this experiment. Box-Behnken design for response surface methodology was used to optimize the technological conditions in the water extraction process of Porphyra haitanensis polysaccharides. Physicochemical properties of Porphyra haitanensis polysaccharides were evaluated: composition of polysaccharides, protein content, FTIR spectrum, effects of concentration, temperature, and pH on the apparent viscosity of PHP solution. Dose-dependent in vitro antioxidant activity using DPPH*, OH*, and ABTS* model systems were evaluated. The degree of hydrolysis during the digestibility of PHPs by artificial human gastric juice was evaluated. Obtained results of antioxidant activity and digestibility suggest the conclusion that PHP have potential prebiotic activity and demonstrate the potential for use in the food and cosmetic industries.

The obtained results were processed statistically and compared with the results of other authors in the literature.

Comments to authors: • Line 166 "were determined in Table 3" - for me sounds better "was presented". We present our results, determined during analysis. • DPPH and ABTS should be with free radical symbol (DPPH*, ABTS*) • Section “3.4.2. PHP composition” should be improved: that was HPLC conditions (what kind of detector, mobile phase was used, flow and temperature, other conditions). It is impossible to repeat experiment according to written description.

Author Response

Dear reviewer:
       Thank you very much for your comments and guidance on this article during your busy schedule. I have revised it based on your revision comments. Please refer to the document sent to you for details.
                                                                                               Yours sincerely,
                                                                                             Mingshuang Dong

Reviewer 2 Report

The manuscript is concerned with antioxidant activity of sulfated polysaccharides in seaweed, and contains interested results. However, the authors have to dissolve several problems for acceptance of Marine Drugs.

(1) The title is unclear. Change the title, for example, "Antioxidant activity of sulfated polysaccharides in seaweed, Porphyra haitanesis".

(2) Abstract and text

It is difficult to know the abbreviations. Before the first appearance of the abbreviations, the whole name should be written.

(3) Change the order of chapters, like to 1. Introduction, 2. Method and materials, 3. Results and Discussion, 4. Conclusions. It is difficult to read the distance between "Results and Discussion" and "Conclusions".

(4) Lines 102-163.

Extraction of sulfated polysaccharides in marine algae is not new and it is difficult to understand the optimization of polysaccharides extraction parts between lines 102 and 163 including Tables 1 and2, and Figure 2. The difference is not so large. Please rewrite these parts more ease understanding.

(5) Table 3

The purity of polysaccharides seems to be low, 65%? After purification, the authors analyze again the sugar analysis (monosaccharide composition).

(6) L269

Why the authors carried out alpha-amylase digestion of PHPs, which is not amylase analogs, a galactosyl polysaccharide. Healthy foods for diet? The precise explanations are needed.

What is FOS?

(7) Small mistakes,

at line 20, Using; line 27, comma; line 39, d; line 40, l; line 242, "respectively were" should be "respectively, were"; line 276, prebiotc should be probiotic.

Author Response

Dear reviewer:

Thank you very much for your comments and guidance on this article during your busy schedule. I have modified it according to your modification comments. The detailed description is as follows:

(1)I have modified it based on your comments, and the title has changed to "Determination of the extraction, physicochemical characterization and digestibility of sulfated polysaccharides in seaweed— Porphyra haitanensis"

(2)Modifications have been made based on the comments。

(3)Because the publishing format of the journal requires this typesetting order, the order of my articles was modified to this order. Before the revision, the order of my articles was consistent with what you said

(4)In this part of the single factor experiment, I conducted a single factor analysis of variance through the SPSS software, and screened out the optimal single factor conditions, and performed subsequent response surface optimization experiments, A steep slope on the 3D graph surface, and densely integrated elliptical contour lines indicate that two factors interact greatly, However, a gentle slope and round contour lines indicate that the interaction between two factors is not significant,The experimental results show that the interaction of the two factors is not significant, while the effect of the three factors is significant,it shows that the result of this optimization model is reliable. I added the response surface 3D diagram in figure 2 for a more intuitive understanding.

 (5) I am very sorry for this part of the content, because my mistakes in writing have caused you unnecessary trouble. This should be 85% instead of 65%. After purification, the experiment of monosaccharide composition was done.

 (6) Because I want to predict its digestibility and ensure that it can safely reach the stomach, because saliva contains a-amylase, a-amylase is selected for the in vitro simulation experiment.

          FOS is Fructooligosaccharide

 (7) The corresponding symbol problem has been revised, but in the question of "line 276, prebiotc should be probiotic", I would like to make a corresponding statement. Prebiotics refer to some that are not digested by the host but can selectively promote the metabolism and Organic substances that multiply and improve the health of the host, so prebiotics are more appropriate here.

I will hand you two documents: one is the revision of the manuscript and the other is the revision of the figure. Please check it.

                                                                                                 Yours sincerely,

                                                                                             Mingshuang Dong

Reviewer 3 Report

The manuscript is an interesting document that includes an extensive study on polysaccharides from Porphyra haitanensis. Specifically, it deals with the determination of an optimized method to extract these polysaccharides and perform subsequent evaluation of their physicochemical properties and digestibility.
Although there are already citations in the literature (e.g. number 13 in the manuscript) about previous similar works on this seaweed, the optimization study on the extraction method through using the Box-Behnken response surface design, as well as the detailed analysis of antioxidant activity, rheological properties, viscosity, component characterization, etc., make, in my opinion, the work described herein an interesting contribution within this field of study.
The experimental section presents broad and consistent data concordant with the discussion previously carried out. In general, the distribution of content is appropriate. facilitating the reading, to my mind. Authors have properly report the experimental procedures and the results obtained. I do not miss any concept related to the general topic that has not been inserted in this manuscript.
The layout and presentation of the manuscript is rather satisfactory, and the amount of references used and their degree of novelty are adequate.
Thus, I recommend to consider this manuscript for publication in its current form.

Author Response

Dear reviewer:

Thank you for your approval of this manuscript. I am extremely grateful for taking your precious time to review my manuscript.

                                                                                               Yours sincerely,

                                                                                             Mingshuang Dong